# Cognitive Models as Simulators:
# Using Cognitive Models to Tap into Implicit Human Feedback

**Ardavan S. Nobandegani** [1 2]   **Thomas R. Shultz** [1 3]   **Irina Rish** [2]

## Abstract

In this work, we substantiate the idea of *cognitive models as simulators*, which is to have AI systems interact with, and collect feedback from, cognitive models instead of humans, thereby making the training process safer, cheaper, and faster. We leverage this idea in the context of learning a fair behavior toward a counterpart exhibiting various emotional states — as implicit human feedback. As a case study, we adopt the Ultimatum game (UG), a canonical task in behavioral and brain sciences for studying fairness. We show that our reinforcement learning (RL) agents learn to exhibit differential, rationally-justified behaviors under various emotional states of their UG counterpart. We discuss the implications of our work for AI and cognitive science research, and its potential for interactive learning with implicit human feedback.

## 1. Introduction

Recent years have witnessed artificial intelligence (AI) systems with remarkable abilities (e.g., Devlin et al., 2018; Goyal et al., 2021), whose success critically depends on having access to huge amounts of training data. Examples include the famous Google BERT language model pre-trained on 800M words from BooksCorpus and $2,500$M words from Wikipedia (Devlin et al., 2018), the DeepMind AlphaGo system trained on over 30M expert moves (Silver et al., 2016), the OpenAI GPT-3 model pre-trained on 300 billion tokens (Brown et al., 2020), and the recent Facebook SEER image recognition model trained on one billion images from Instagram photos (Goyal et al., 2021).

---
[1]Department of Psychology, McGill University, Montreal, Canada [2]Mila – Quebec AI Institute, Montreal, Canada [3]School of Computer Science, McGill University, Montreal, Canada. Correspondence to: Ardavan S. Nobandegani <ardavan.salehinobandegani@mcgill.ca>.

Interactive Learning with Implicit Human Feedback Workshop at ICML 2023, Honolulu, Hawaii, USA.

Similarly, in reinforcement learning (RL), agents need to have many interactions with their environment to collect feedback in the form of rewards (Sutton & Barto, 2018). This is especially challenging in settings where the environment consists of human agents, resulting in these interactions being expensive, time-consuming, and potentially unsafe — thus exacerbating the training process. Could we instead use cognitive models, as a *proxy* for humans, to address this issue?

In this work, we substantiate the idea of *cognitive models as simulators*, which is to have AI systems interact with, and collect feedback from, cognitive models instead of humans, thereby making the training process safer, cheaper, and faster. Focusing on emotions as a form of implicit human feedback, we leverage this idea in the context of learning a fair behavior toward a counterpart exhibiting various emotional states.

A substantial body of work in emotion research shows that people display emotions as a means of communication (e.g., Tronick, 1989; Parr et al., 2005; Planalp et al., 2006), serving as implicit feedback that clues others as to how they should regulate their behavior. For example, when someone we care about displays sadness, we are likely to know that we need to provide support (Planalp et al., 2006).

As a case study, we adopt the Ultimatum game (UG), a canonical task in behavioral and brain sciences for studying fairness (e.g., Sanfey, 2009; Battigalli et al., 2015; Vavra et al., 2018; Sanfey et al., 2003; Xiang et al., 2013; Chang & Sanfey, 2013). As we show, our RL agents learn to exhibit differential, rationally-justified behaviors under various emotional states of a simulated UG Responder (see Section 2 for an explanation of how UG works).

We begin by describing UG and presenting an overview of the relevant psychological findings on the role of emotions in UG (Section 2). We then discuss in Section 3 a cognitive model of UG Responder under a variety of emotional states (Lizotte et al., 2021; Nobandegani et al., 2020), and subsequently present our RL training results under various UG Responder's emotional states (Section 4). We then discuss relevant past work (Section 5), and conclude by discussing the implications of our work for AI and cognitive science re-

search, and its potential for interactive learning with implicit human feedback (Section 6).

## 2. UG and the Role of Emotions in UG

The Ultimatum game (UG; Güth et al., 1982) is a canonical task for studying fairness, and has been extensively studied in psychology (e.g., Sanfey, 2009; Battigalli et al., 2015; Vavra et al., 2018), neuroscience (Sanfey et al., 2003; Xiang et al., 2013; Chang & Sanfey, 2013), philosophy (Guala, 2008), and behavioral economics (e.g., Güth et al., 1982; Thaler, 1988; Camerer & Thaler, 1995; Fehr & Schmidt, 1999; Sutter et al., 2003; Camerer & Fehr, 2006). UG has a simple design: Two players, Proposer and Responder, must agree on how to split a sum of money. Proposer makes an offer. If Responder accepts, the deal goes through; if Responder rejects, neither player gets anything. In both cases, the game is over.

An extensive body of empirical work has established that UG Proposers predominantly respect fairness by offering about 50% of the endowed amount, and that this split is almost invariably accepted by UG Responders (see Camerer, 2011). Relatedly, UG Responders often reject offers below 30%, presumably as retaliation for being treated unfairly (Güth et al., 1982; Thaler, 1988; Güth & Tietz, 1990; Bolton & Zwick, 1995; Nowak et al., 2000; Camerer & Fehr, 2006).

Substantial empirical work has revealed that induced emotions strongly affect UG Responder's accept/reject behavior, with positive emotions increasing the chance of low offers being accepted (e.g., Riepl et al., 2016; Andrade & Ariely, 2009), and negative emotions decreasing the chance of low offers being accepted (e.g., Bonini et al., 2011; Harlé & Sanfey, 2010; Liu et al., 2016; Moretti & Di Pellegrino, 2010; Vargas et al., 2019). Experimentally, these emotions are often induced by a movie clip or recall task.

## 3. A Cognitive Model of UG Responder

Recently, Nobandegani et al. (2020) presented a cognitive model of UG Responder, called *sample-based expected utility* (SbEU). SbEU provides a unified account of several disparate empirical findings in UG (i.e., the effects of expectation, competition, and time pressure on UG Responder), and also explains the effect of a wide range of emotions on UG Responder (Lizotte et al., 2021).

Nobandegani et al.'s cognitive model rests on two main assumptions. First, UG Responder uses SbEU to estimate the expected-utility gap between their expectation and the offer, i.e., $\mathbb{E}[u(\text{offer}) - u(\text{expectation})]$, where $u(\cdot)$ denotes Responder's utility function. If this estimate is positive — indicating that the offer made is, on average, higher than Responder's expectation — Responder accepts the offer;

otherwise, Responder rejects the offer. This assumption is supported by substantial empirical evidence showing that Responder's expectation serves as a reference point for subjective valuation of offers (Sanfey, 2009; Battigalli et al., 2015; Vavra et al., 2018; Xiang et al., 2013; Chang & Sanfey, 2013).

The second assumption is that negative emotions elevate loss-aversion while positive emotions lower loss-aversion (Lizotte et al., 2021). Again, this assumptions is supported by mounting empirical evidence (e.g., De Martino et al., 2010; Sokol-Hessner et al., 2015; 2009) suggesting that emotions modulate loss-aversion — the tendency to overweight losses as compared to gains (Kahneman & Tverskey, 1979).

Concretely, SbEU assumes that an agent estimates expected utility:

$$\mathbb{E}[u(o)] = \int p(o)u(o)do, \tag{1}$$

using self-normalized importance sampling (Nobandegani et al., 2018; Nobandegani & Shultz, 2020b;c), with its importance distribution $q^*$ aiming to optimally minimize mean-squared error (MSE):

$$\hat{E} = \frac{\sum_{i=1}^{s} w_i u(o_i)}{\sum_{j=1}^{s} w_j}, \quad \forall i : o_i \sim q^*, \; w_i = \frac{p(o_i)}{q^*(o_i)}, \tag{2}$$

$$q^*(o) \propto p(o)|u(o)|\sqrt{\frac{1 + |u(o)|\sqrt{s}}{|u(o)|\sqrt{s}}}. \tag{3}$$

MSE is a standard measure of estimation quality, widely used in decision theory and mathematical statistics (Poor, 2013). In Eqs. (1-3), $o$ denotes an outcome of a risky gamble, $p(o)$ the objective probability of outcome $o$, $u(o)$ the subjective utility of outcome $o$, $\hat{E}$ the importance-sampling estimate of expected utility given in Eq. (1), $q^*$ the importance-sampling distribution, $o_i$ an outcome randomly sampled from $q^*$, and $s$ the number of samples drawn from $q^*$.

SbEU has so far explained a broad range of empirical findings in human decision-making, e.g., the fourfold patterns of risk preferences in both outcome probability and outcome magnitude (Nobandegani et al., 2018), risky decoy and violation of betweenness (Nobandegani et al., 2019c), violation of stochastic dominance (Xia et al., 2022), violation of cumulative independence (Cao et al., 2022), the three contextual effects of similarity, attraction, and compromise (da Silva Castanheira et al., 2019), the Allais, St. Petersburg, and Ellsberg paradoxes (Nobandegani & Shultz, 2020b;c; Nobandegani et al., 2021), cooperation in Prisoner's Dilemma (Nobandegani et al., 2019a), and human coordination behavior in coordination games (Nobandegani & Shultz, 2020a).

# 4. Training RL Agents in UG

In this section, we substantiate the idea of *cognitive models as simulators* in the context of moral decision-making (Haidt, 2007; Lapsley, 2018), by having RL agents learn about fairness through interacting with a cognitive model of UG Responder (Nobandegani et al., 2020), as a proxy for human Responders, thereby making their training process both less costly and faster.

To train RL Proposers, we leverage the broad framework of multi-armed bandits in reinforcement learning (Katehakis & Veinott, 1987; Gittins, 1979), and adopt the well-known Thompson Sampling method (Thompson, 1933). Specifically, we assume that RL Proposer should decide what percentage of the total money $T$ they are willing to offer to SbEU Responder. For ease of analysis, here we assume that RL Proposer chooses between a finite set of actions: $\mathcal{A} = \{0, \frac{T}{10}, \frac{2T}{10}, \cdots, \frac{9T}{10}, T\}$.

---

**Algorithm 1** Thompson Sampling for UG Proposer

---

    **Initialize**. $\forall a \in \mathcal{A}$: $S_a = 0$ and $F_a = 0$
1:  **for** $i = 1, \ldots, N$
2:  $\forall a \in \mathcal{A}$ compute:
    $s_a = u(T - a)\beta_a, \quad \beta_a \sim \text{Beta}(S_a + 1, F_a + 1)$
3:  $a^* = \arg\max_a \ s_a$
4:  Offer $a^*$ to SbEU Responder
5:  **if** SbEU Responder accepts the offer **then**
6:      $S_{a^*} = S_{a^*} + 1$
7:  **else**
8:      $F_{a^*} = F_{a^*} + 1$
9:  **end if**
10: **end for**

---

In reinforcement learning terminology, RL Proposer learns, through trial and error while striking a balance between exploration and exploitation, which action $a \in \mathcal{A}$ yields the highest expected reward. Here, we train RL Proposers using Thompson Sampling, a well-established method in the RL literature enjoying near-optimality guarantees (Agrawal & Goyal, 2012; 2013); see Algorithm 1.

Algorithm 1 can be described in simple terms as follows. At the start, i.e., prior to any learning, the number of times an offer $a \in \mathcal{A}$ is so far accepted, $S_a$ (S for success), and the number of times it is rejected, $F_a$ (F for failure), are both set to zero. In each trial (for a total of $N$ trials), an estimate of expected reward for each offer $a \in \mathcal{A}$ is computed by sampling from the corresponding distribution (Line 2), and the offer with the highest expected reward estimate $a^*$ (Line 3) is then chosen by Proposer to be offered to SbEU Responder (Line 4). If this offer is accepted by SbEU Responder, the $S_a$ parameter for that offer is incremented by one (Line 6); if rejected, the $F_a$ parameter for that offer is instead incremented by one (Line 8). In Algorithm 1, $T$ is the total

amount of money to be split between Proposer and Responder, $u(\cdot)$ is the subjective utility function of Responder, and Beta$(\cdot, \cdot)$ is the Beta distribution.

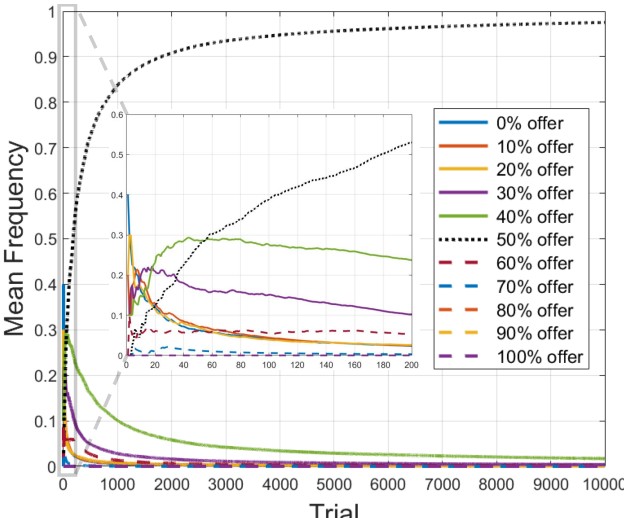

*Figure 1.* Mean frequency of RL Proposer's offers. The $y$-axis indicates the mean frequency of each offer made by RL Proposer to SbEU Responder up to current trial ($x$-axis), averaged over 10 RL Proposers. SbEU Responder is in a neutral emotional state. As a visual aid, the dynamics for the first 200 trials are provided in a smaller plot, located at the center.

In Figure 1, we simulate 10 RL Proposers, and report the mean frequency of an offer being made to SbEU Responder over the past trials, for a total of $N = 10,000$ trials. As can be seen, exercising a balance between exploration and exploitation, RL Proposers eventually arrive at the decision that they should be making a fair offer to SbEU Responder, i.e., to split the total sum $T$ equally between themselves and Responder.

## 4.1. RL Proposer Meets Emotional Responder

Tapping into emotions as a form of implicit human feedback, here we bridge between the idea of *cognitive models as simulators* and emotion research, by letting AI systems interact with a cognitive model of people experiencing various emotional states. Specifically, we pursue this idea in the context of UG, and have RL Proposers interact with SbEU Responders experiencing positive and negative emotional states — as implicit human feedback.

A wealth of empirical research has revealed that the effect of emotions on human decision-making is both substantial and systematic (for reviews see, e.g., Phelps et al., 2014; Lerner et al., 2015). More specifically, in the context of UG, a growing body of empirical studies have shown that induced emotions strongly affect UG Responder's behavior, with

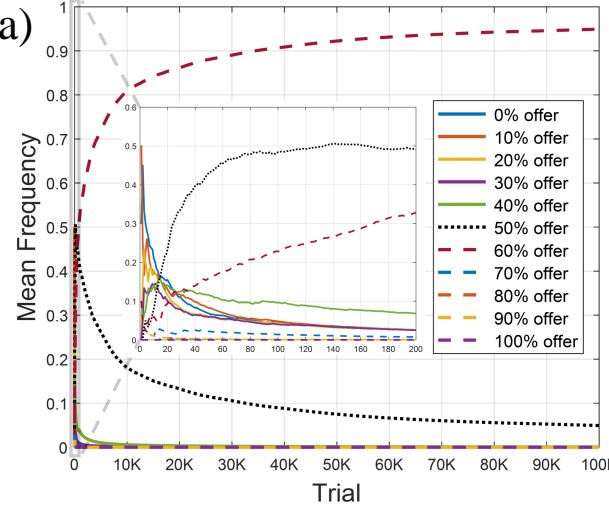

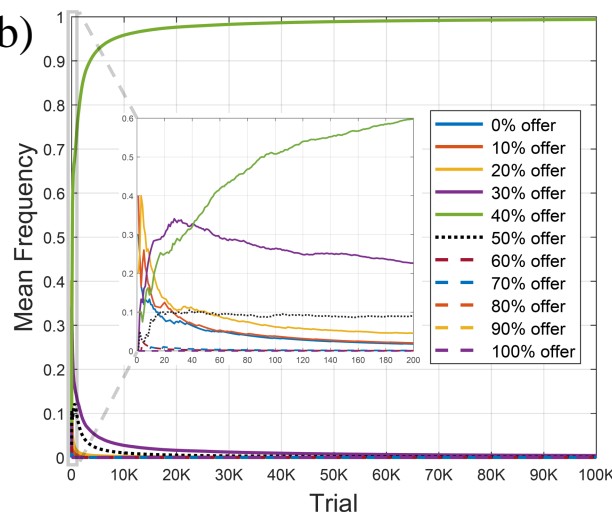

*Figure 2.* Mean frequency of RL Proposer's offers. The $y$-axis indicates the mean frequency of each offer made by RL Proposer to SbEU Responder up to current trial ($x$-axis), averaged over 10 RL Proposers. In **(a)** SbEU Responder is under a negative emotional state, while in **(b)** SbEU Responder is under a positive emotional state. As a visual aid, in each subplot, the dynamics for the first 200 trials are provided in a smaller plot, located at the center.

positive emotions (e.g., happiness) increasing the chance of low offers being accepted (e.g., Riepl et al., 2016; Andrade & Ariely, 2009), and negative emotions (e.g., disgust, anger, and sadness) decreasing the chance of low offers being accepted (e.g., Bonini et al., 2011; Harlé & Sanfey, 2010; Liu et al., 2016; Moretti & Di Pellegrino, 2010; Vargas et al., 2019). Hence, it would be rational for UG Proposer (from the perspective of maximizing their expected reward) to make larger offers to Responders experiencing negative emotions, and, conversely, to make smaller offers to Responders experiencing positive emotions.

Interestingly, under the broad and empirically well-supported assumption that emotions modulate loss-aversion (e.g., De Martino et al., 2010; Sokol-Hessner et al., 2015; 2009), Nobandegani et al.'s (2020) SbEU model explains the effect of a wide range of emotions on human UG Responder (Lizotte et al., 2021). Next, we train RL Proposers, using Thompson Sampling (see Algorithm 1), to learn how to interact with SbEU Responders experiencing positive or negative emotional states.

In Figure 2, we simulate 10 RL Proposers, and report the mean frequency of an offer being made to SbEU Responder over the past trials, for a total of $N = 100,000$ trials. In Figure 2(a), SbEU Responder is under a negative emotional state, and, in Figure 2(b), SbEU Responder is under a positive emotional state. As can be seen, RL Proposers eventually arrive at the decision that they should be making a larger offer ($60\%$) when Responder is experiencing a negative emotional state (Figure 2(a)), and, conversely, should be making a smaller offer ($40\%$) when Responder is experiencing a positive emotional state (Figure 2(b)).

As such, taking into account the emotional state of their UG counterpart, which serves as an important source of implicit human feedback, RL proposers learn to exhibit differential, rationally-justified behaviors under various emotional states of their UG Responder, i.e., neutral (Figure 1), negative (Figure 2(a)), and positive (Figure 2(b)).

## 5. Related Work

Past work has leveraged data generated by cognitive models, and more broadly, models of human behavior, to train AI systems (Bourgin et al., 2019; Carroll et al., 2019; Trafton et al., 2020; Zhang et al., 2021; Sense et al., 2022; Hu et al., 2022).

Bourgin et al. (2019) focused on the problem of predicting human decisions when choosing between risky gambles, and leveraged the data generated by a cognitive model of human decision-making (BEAST; Erev et al., 2017) to train a neural-network model achieving state-of-the-art performance in predicting human risky decision-making. Specifically, Bourgin et al. (2019) used the synthetic data generated by BEAST to pretrain their neural-network model, allowing it to start off from a good initialization.

Trafton et al. (2020) used an extension of a well-known cognitive model, (ACT-R; Anderson et al., 2004), to generate synthetic data which would then be used to train a deep neural-network model predicting human actions in a supervisory control task. Their trained neural-network showed superior predictive performance compared to a classifier trained solely on (limited) empirical data.

Sense et al. (2022) used a cognitive model of human memory, (PPE; Jastrzembski et al., 2006) to engineer timing-related input features for a gradient-boosted decision trees (GBDT) model. The resulting PPE-enhanced GBDT outperformed the default GBDT, especially under conditions in which limited data were available for training.

Carroll et al. (2019) focused on the problem of human-AI coordination, in an environment based on the popular game *Overcooked*. Carroll et al. evaluated the performance of agents trained via self-play and population-based training, and showed that these agents performed well when paired with themselves, but when paired with a human model, they were significantly worse than agents trained to play with the human model. Carroll et al. developed their human model by behavioral cloning (Bain & Sammut, 1995).

In subsequent work, Hu et al. (2022) considered the problem of human-AI coordination in partially observed environments, and developed a three-step algorithm that achieved strong performance in coordinating with real humans in the Hanabi benchmark (Bard et al., 2020). Hu et al. (2022) used a regularized search algorithm and behavioral cloning to produce a human model and then integrated a policy regularization method into reinforcement learning to train a human-like best response to the human model.

Perhaps the closest work to our work is a short, position paper by Zhang et al. (2021) in the domain of human-computer interaction. Zhang et al. proposed using cognitive models to pretrain RL agents before they are applied to real human users, thereby endowing those RL agents with a good initial policy — dubbed *warm start* RL agents. Zhang et al. reviewed two case studies; one was a mobile notification app motivating physical exercise in a human user, and the other was a driving assist app that helps human drivers to keep lanes. As a model of human user with which the RL agent (i.e., the app) interacts, Zhang et al. used a dynamic Bayesian network in the former case, and ACT-R, in the latter.

To our knowledge, ours is the first interactive learning work that uses cognitive models to train RL agents that tap into human's display of emotions, as an important source of implicit human feedback.

## 6. Discussion

To achieve desirable performance, current AI systems often require huge amounts of training data. This is especially problematic in domains where collecting data is both expensive and time-consuming, e.g., where AI systems require many interactions with humans, collecting feedback from them. In this work, we substantiate the idea of *cognitive models as simulators*, which is to have AI systems interact with, and collect feedback from, cognitive models as

a proxy for humans, thereby making their training process safer, cheaper, and faster.

Focusing on emotions as an important source of implicit human feedback, we leverage the idea of *cognitive models as simulators* in the context of learning a fair behavior toward a counterpart exhibiting various emotional states. A substantial body of work in emotion research shows that people display emotions as a means of communication (e.g., Tronick, 1989; Parr et al., 2005; Planalp et al., 2006), serving as implicit feedback that clues others into how they should regulate their behavior.

As a case study, we adopt the Ultimatum game (UG), a canonical task in behavioral and brain sciences for studying fairness (e.g., Sanfey, 2009; Battigalli et al., 2015; Vavra et al., 2018; Sanfey et al., 2003; Xiang et al., 2013; Chang & Sanfey, 2013). As a cognitive model, we use sample-based expected utility (SbEU), a psychological model that explains a wide range of empirical findings on UG Responders (Nobandegani et al., 2020; Lizotte et al., 2021). As an AI system, we train RL Proposers using Thompson Sampling, a well-known method in the multi-armed bandits literature, enjoying near-optimality guarantees (Agrawal & Goyal, 2012; 2013). As we show, our RL agents learn to exhibit differential, rationally-justified behaviors under various emotional states of their simulated UG Responder, making larger offers when Responder is more likely to reject low offers (due to experiencing negative emotions) and, conversely, making smaller offers when Responder is less likely to reject low offers (due to experiencing positive emotions).

Although here we focused on the three broad categories of neutral, negative, and positive emotions, SbEU allows for simulating UG Responder under various nuanced emotional states (e.g., sadness, anger, disgust, happiness; Lizotte et al., 2021), thus permitting a more specialized training of RL Proposers.

Recent success stories in AI, e.g., AlphaGo and particularly self-play (Silver et al., 2016; 2017), clearly demonstrate the significant role that having access to a simulator of the environment would play in efficient training of AI systems. The idea of *cognitive models as simulators* substantiated in this work is yet another step in the direction of leveraging simulators of the environment — by using cognitive models as a proxy for people — in the service of making the training of AI systems safer, cheaper and faster. As such, the idea of *cognitive models as simulators* presents an important way for computational cognitive science to contribute to AI.

Interestingly, as cognitive models allow individual-level modeling of humans, taking into account individual-level differences among people (e.g., their emotional states), the idea of *cognitive models as simulator* paves the way for personalized training of AI agents interacting with human

users.

Additionally, as cognitive process models serve as proposals for a causal generative model of behavior, they could be effectively used to simulate interventions and counterfactuals, all of which improves generalization of AI training.

Although here we presented the idea of *cognitive models as simulators* as a way of making the training of AI systems more efficient, it could also be seen as a broad *cognitive* framework for how people might be choosing their strategies in multi-agent environments by mentalizing about other agents. As such, the idea of *cognitive models as simulators* could potentially serve as a broad framework for theorizing about, and mathematically identifying, mental processes by which people choose their strategies when interacting with other agents. Hence, this "cognitive" reconceptualization of *cognitive models as simulators* has potential to make contributions to computational cognitive science.

Additionally, a strong reading of this cognitive reconceptualization takes the AI systems learning from interacting with mental models as a proposal for how people might be choosing their strategy in multi-agent environments, thus presenting an important way for AI to contribute to computational cognitive science.

From this perspective, the Thompson Sampling algorithm presented in Section 4 for training RL Proposers could serve as a process-level proposal for how human Proposers might be choosing their offer: by simulating UG Responder using a mental model of UG Responder and learning from mentally interacting with that model, here implemented by SbEU (Nobandegani et al., 2020). Nonetheless, human Proposers might be using a much simpler mental model of their human Responder as compared to SbEU, and would presumably start with much stronger prior beliefs (i.e., inductive biases) about the expected reward of each of their strategies — instead of the uniformly distributed $\text{Beta}(1, 1)$ prior used in Algorithm 1. Future work should more extensively investigate this process-level proposal.

Also, this cognitive reconceptualization is consistent with substantial work on both people's intuitive psychology and human strategic decision-making (e.g., Jern et al., 2017; Jara-Ettinger et al., 2016; Nagel, 1995; Baker et al., 2009; Camerer et al., 2004), broadly assuming that people have a mental model of other agents and use that model to both interpret other agents' behavior and decide how to behave when interacting with those agents.

To our knowledge, ours is the first interactive learning work that uses cognitive models to train RL agents that tap into human's display of emotions, as an important source of implicit human feedback. We see our work as a step in the direction of developing AI systems that regulate their interaction with humans depending on the emotional state of their human counterparts.

## Acknowledgements

This research was supported in part by an operating grant to TRS from the Natural Sciences and Engineering Research Council of Canada (NSERC). ASN and IR acknowledge the support from Canada CIFAR AI Chair program and from the Canada Excellence Research Chairs (CERC) program.

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
