# OpenReview forum: "Cognitive Models as Simulators: Using Cognitive Models to Tap into Implicit Human Feedback"
_ICML.cc/2023/Workshop/ILHF — ILHF Workshop ICML 2023_

### Official Review · Reviewer_scBQ · 2023-06-15
**The paper presents an interesting and well-motivated empirical study, but the setting considered is very simplistic**

**Rating:** 6
**Confidence:** 4

**Review:**

Review:
---
The paper proposes using offline RL/bandits to learn agents that behave fairly towards differently-behaving humans. The setting considered is the Ultimatum game, where the RL agent chooses how to divide a large sum between itself and a human model, where the human model may behave differently based on different emotional states.

I think the paper is well-written and motivated well, as interacting with cognitive models will be much less expensive than real humans. However, the exact experiment is very simple and it is unclear if the findings generalize to more realistic settings. First, the environment is a bandit problem rather than an RL one, without any context and a discrete set of 11 actions. Second, the paper only considers humans as exhibiting only three different emotional states. It would be helpful if the paper offered some discussion on how a corresponding experiment could be set up in a more realistic setting.

Minor Comments:
---
(1) To my knowledge, the paper does not describe how the different emotional states are modeled under the SbEU model that the paper assumes. It is done via changing the utility function?

(2) The plots show a large timescale of interactions, where the behavior of the bandit agents appears to converge in under 10% of the interactions used. It could be useful to reduce the timescale of the plots.

---

### Official Review · Reviewer_U9QN · 2023-06-16
**Good addition to workshop but needs some revision**

**Rating:** 6
**Confidence:** 4

**Review:**

This work proposes using "cognitive models", i.e. models of human behaviour as observed through existing or carefully performed experiments, as simulators for training RL agents. This way, the agent would learn to interact within an environment that responds similarly to how a human would, for example with respect to varying emotional states. This would also eliminate the need for collecting a lot of data during training.

As far as I know, the idea of using cognitive models as simulators for training RL agents is novel and for that reason I think this paper would be a good fit for this workshop. I do have some concerns however:

1) The cognitive models themselves are built out of existing data to replicate the behaviour of humans in very limited settings that have been observed in controlled experiments. Outside of these settings, we cannot know if the models are good models of human behaviour or not and when training an RL agent, we are likely to venture into data distributions that were not observed in controlled settings. As a result, it's highly dubious that these cognitive models would still be useful as simulators. I would love to see a discussion of this in the paper and how modelling issues could be addressed or different models of the same behaviour could exist.

2) Section 2 is missing a lot of pertinent technical details that would help in understanding how the UG Responder is modeled. For example, where does the utility function come from? How is emotion of the Responder modeled?

3) Since the example given is a bandit problem, I think the proposer shouldn't be called "RL Proposer" or "RL agent" (Section 4) as that implies multiple decision-making steps. The full RL problem can be mentioned as the next step for the work.

---

### Decision · Program_Chairs · 2023-06-20

Accept